https://doi.org/10.1038/s41467-021-25598-0　　**OPEN**

# Directed evolution of prenylated FMN-dependent Fdc supports efficient in vivo isobutene production

Annica Saaret [1], Benoît Villiers[2✉], François Stricher [2], Macha Anissimova [2], Mélodie Cadillon[2], Reynard Spiess[1], Sam Hay [1] & David Leys [1✉]

Isobutene is a high value gaseous alkene used as fuel additive and a chemical building block. As an alternative to fossil fuel derived isobutene, we here develop a modified mevalonate pathway for the production of isobutene from glucose in vivo. The final step in the pathway consists of the decarboxylation of 3-methylcrotonic acid, catalysed by an evolved ferulic acid decarboxylase (Fdc) enzyme. Fdc belongs to the prFMN-dependent UbiD enzyme family that catalyses reversible decarboxylation of (hetero)aromatic acids or acrylic acids with extended conjugation. Following a screen of an Fdc library for inherent 3-methylcrotonic acid decarboxylase activity, directed evolution yields variants with up to an 80-fold increase in activity. Crystal structures of the evolved variants reveal that changes in the substrate binding pocket are responsible for increased selectivity. Solution and computational studies suggest that isobutene cycloelimination is rate limiting and strictly dependent on presence of the 3-methyl group.

---

[1] Department of Chemistry, Manchester Institute for Biotechnology, University of Manchester, Manchester, UK. [2] Global Bioenergies, Evry-Courcouronnes, France.
✉email: benoit.villiers@global-bioenergies.com; david.leys@manchester.ac.uk

The irrefutable harmful environmental effects and depleting reserves of fossil fuels have powered an extensive amount of research to seek sustainable alternatives for the production of petrochemicals, including the gaseous alkene isobutene[1–3]. Due to the favourable reactivity, isobutene is widely used as a building block for fuel additives, rubber, plastic and a broad range of fine chemicals. Over 10 million tons of isobutene are produced every year, primarily by steam cracking crude oil. Low levels of microbial production of isobutene were first detected in the 1980s[4–6]. More recently, isobutene production via a modified mevalonate (MVA) pathway using mevalonate diphosphate decarboxylase (MVD) to decarboxylate 3-hydroxyisovaleric acid was reported[7] and subsequently patented[8] (Fig. 1A). Further studies highlighted a more efficient route using mevalonate-3-kinase (M3K, *Picrophilus torridus*) that catalyses isobutene formation through an unstable phosphorylated intermediate[9,10]. The highest reported whole-cell isobutene production rate of 507 pmol min$^{-1}$ g cells$^{-1}$ was reached using *E. coli* engineered with M3K, however, this remains about 10$^5$-fold lower than is economically viable[7,9]. The slow conversion could be surpassed by an alternative route, such as the more direct conversion of methylcrotonyl-CoA to isobutene through a combination of a thioesterase with a non-oxidative decarboxylase. The prenylated flavin (prFMN)-dependent ferulic acid decarboxylases (Fdc) catalyse reversible non-oxidative (de) carboxylation of a range of acrylic acids with extended conjugation[11–14]. Recently, a reversible 1,3-dipolar cycloaddition mechanism was conclusively shown to underpin catalysis in *Aspergillus niger* Fdc (*An*Fdc)[11,15–17]. First, the cycloaddition of the substrate results in cycloadduct **Int1** (Fig. 1B). Decarboxylation occurs concomitantly with ring opening to form **Int2**. Following the exchange of $CO_2$ with E282, protonation by E282 results in cycloadduct **Int3** that releases the product through cycloelimination. Cycloadduct strain, mediated by a clash between the substrate R group and enzyme residues is key in ensuring reversible 1,3-dipolar cycloaddition[17]. Recent studies have shown rational engineering of *An*Fdc can expand substrate scope to include aromatic substrates such as naphthoic acid[18]. Crotonic

acid was found to inhibit *An*Fdc by apparently irreversibly binding to the prFMN cofactor[17]. However, acrylic acid substrates lacking extended conjugation have rarely been reported in the wider UbiD enzyme family. Arguably, the natural UbiD substrate closest to 3-methylcrotonic acid is trans-anhydromevalonate 5-phosphate (tAHMP), which is decarboxylated by a UbiD-decarboxylase from a hyperthermophilic archaeon *Aeropyrum pernix* in an alternative mevalonate pathway[19]. Both 3-methylcrotonic acid and tAHMP contain a secondary beta carbon and lack extended conjugation, however, the phosphate group in tAHMP may facilitate strain manipulation in cycloadduct intermediates. A recent communication describes a pathway for the production of butadiene in *E. coli* where the decarboxylation of *cis,cis*-muconic acid is catalysed by Fdc from *Saccharomyces cerevisiae* enhanced by rational design[20].

Herein, we report on discovery and optimization through directed evolution of Fdc decarboxylation activity with 3-methylcrotonic acid to produce isobutene. We seek to understand how a substrate lacking extended conjugation and bulk can be decarboxylated by Fdc, especially in view of the fact crotonic acid acts as an inhibitor of prFMN. We discuss the structural basis for an increase in activity and selectivity in *Trichoderma atroviride* Fdc (*Ta*Fdc) evolved by directed evolution. Surprisingly, the optimized variants remain unable to decarboxylate crotonic acid, suggesting that in the case of the substrate 3-methylcrotonic acid the single additional methyl group plays a key role in the cycloelimination process. Computational studies are used to rationalize the effect of the 3-methyl substitution on product formation.

## Results and discussion

**Initial screening of Fdc homologues.** Initial in vivo screening tested 15 UbiD homologues co-expressed with UbiX (*E. coli* K-12) in *E. coli* for conversion of 3-methylcrotonic acid into isobutene as detected by gas chromatography. *Ta*Fdc exhibited over twice the isobutene production compared to other homologues

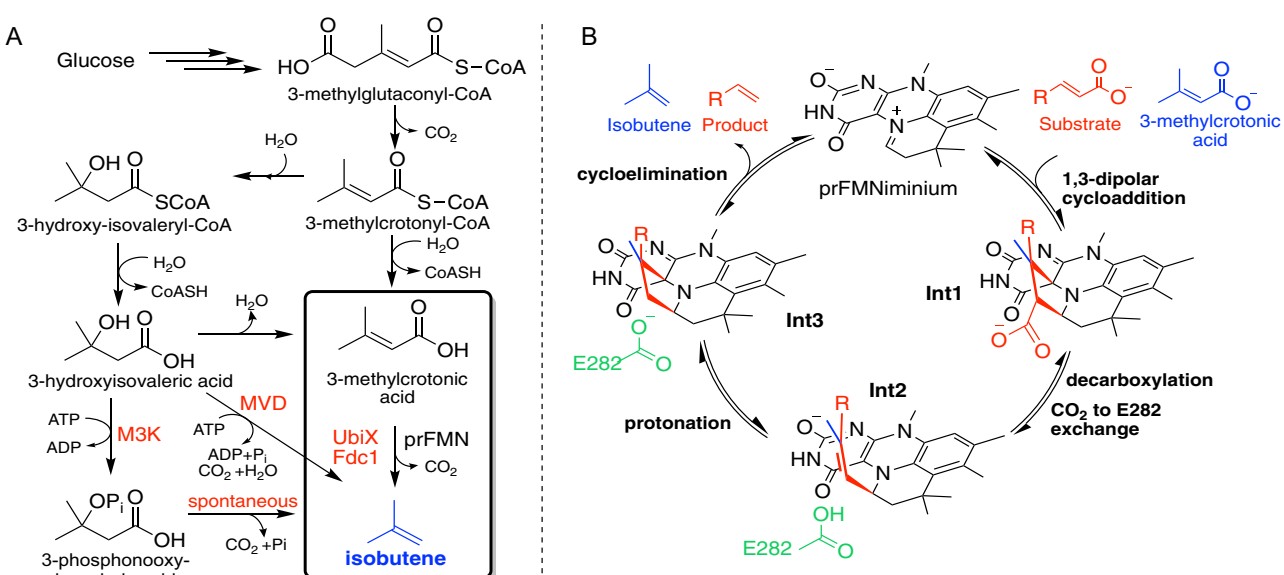

**Fig. 1 Isobutene production via the modified mevalonate pathway utilizing Fdc for the last step. A** Routes to isobutene via modified mevalonate pathways. Previously published mevalonate diphosphate decarboxylase (MVD)[7] and mevalonate-3-kinase (M3K)[9] produce isobutene via 3-hydroxyisovaleric acid. Fdc1, co-expressed with UbiX, catalyses the decarboxylation of 3-methylcrotonic acid to give isobutene. **B** Fdc decarboxylation reaction mechanism with 3-methylcrotonic acid in blue and common Fdc substrates with a conjugated R-group in red. First, the 1,3-dipolar cycloaddition of the substrate to prFMN$^{iminium}$ leads to the first pyrrolidine cycloadduct, **Int1**. Decarboxylation and ring-opening forms the noncyclic alkene adduct **Int2**. Protonation by a conserved glutamic acid residue yields the second pyrrolidine cycloadduct **Int3** followed by cycloelimination to give the product.

**Table 1 Summary table of *An*Fdc and *Ta*Fdc variants.**

| | *An*Fdc | *An*FdcI | *An*FdcII | *Ta*Fdc | *Ta*FdcI[a] | *Ta*FdcV[a] | *Ta*FdcII |
|---|---|---|---|---|---|---|---|
| Mutations | WT | T395M | T395M R435P P438W | WT | T405M | E25N N31G G305A D351R K377H P402V F404Y T405M T429A V445P Q448W | F404Y T405M V445P Q448W |
| Crystal structure | 4ZA4 | 7NF3 | 7NF4 | 7NEY | No crystal structure | 7NF1 | No crystal structure |

[a]Product of directed evolution.

(Supplementary Table 1) and a directed evolution approach was taken to generate a variant of *Ta*Fdc with superior isobutene production activity and selectivity for 3-methylcrotonic acid over cinnamic acid (schematically presented in Supplementary Fig. 1). *Ta*FdcI, with a T405M mutation, was the first variant with a considerable increase in isobutene. *Ta*FdcV generated by 4 rounds of evolution has 11 mutations: E25N, N31G, G305A, D351R, K377H, P402V, F404Y, T405M, T429A, V445P and Q448W (Table 1).

**Characterization of *Ta*Fdc and *Ta*FdcV**. *Ta*Fdc wild-type and *Ta*FdcV with an N-terminal hexa-histidine tag were co-expressed with *E. coli* K-12 UbiX in *E. coli* and purified with Ni-NTA resin. UV–Vis spectra of both purified proteins exhibit a distinct peak at 380 nm, thought to correspond to the cofactor active form prFMN$^{iminium}$ (Supplementary Fig. 2A)[16]. ESI-MS confirmed the presence of prFMN$^{iminium}$ in both enzyme variants (Supplementary Fig. 2B and C). The shape of the 380 nm peak and cofactor content (assessed by the ratio of absorbances at 280 and 380 nm) varied from batch to batch.

*Ta*Fdc showed decarboxylation activity with cinnamic and sorbic acid, with rates $k_{obs} = 7.2 \pm 0.3$ and $3.2 \pm 0.3$ s$^{-1}$, respectively (reported for a batch with a 380:280 nm ratio of 0.067). These values are comparable to those reported for *An*Fdc[11,16]. In contrast, the *Ta*FdcV variant showed compromised activity with sorbic acid ($k_{obs} = 0.33 \pm 0.03$ s$^{-1}$) and no activity was detected with cinnamic acid. When exposed to light, *Ta*Fdc sorbic acid decarboxylation activity steadily deteriorates with a half-life of 1 h compared to enzyme stored in dark (Supplementary Fig. 2H). This is consistent with Fdc light-sensitivity as described previously[17]. Upon irradiation with a 405 nm LED lamp, the characteristic 380 nm peak in the UV–visible absorbance spectra of *Ta*Fdc and *Ta*FdcV irreversibly splits to peaks at 365 and 425 nm (Supplementary Fig. 2I and J).

Incubation of both *Ta*Fdc and *Ta*FdcV with 3-methylcrotonic acid triggered a change in the protein UV–Vis spectrum to reveal peaks at 340 and 425 nm, suggestive of a covalent substrate:prFMN adduct accumulating under turnover conditions. Following a desalting step, the spectrum returns to the as-isolated 380 nm single feature, confirming that a long-lived, inhibitory covalent complex with 3-methylcrotonic acid is not formed (Supplementary Fig. 2D and E). Incubation of 80 μmol *Ta*FdcV with 10 mM 3-methylcrotonic acid led to a complete shift in the corresponding UV–Vis spectrum. In contrast, the wild-type *Ta*Fdc required prolonged incubation with 50 mM 3-methylcrotonic acid to achieve full spectral conversion, suggesting a substantially higher $K_D$ and/or adduct formation rate for the wild-type enzyme. An ESI-MS spectrum of the desalted sample showed peaks corresponding to both prFMN$^{iminium}$ and a putative **Int3** prFMN cycloadduct with 3-methylcrotonic acid (Supplementary Fig. 3). This may be due to a small proportion of 3-methylcrotonic acid remains bound to prFMN as **Int3**, suggesting **Int3** elimination is the rate-limiting step, or that a proportion of the **Int3** species has irreversibly isomerized to a more stable conformation.

In order to assess the scope for activity with acrylic acids lacking extended conjugation, *Ta*Fdc and *Ta*FdcV were incubated with *trans*-2-pentenoic and *trans*-2-hexenoic acid, compounds that have previously been reported to undergo some *An*Fdc-mediated decarboxylation[11]. UV–Vis absorbance spectra indicated that *Ta*Fdc bound both acids (Supplementary Fig. 2F), whereas the *Ta*FdcV variant preferred the smaller pentenoic acid and required higher concentrations to fully bind hexenoic acid (Supplementary Fig. 2G). In contrast to samples incubated with 3-methylcrotonic acid, the UV–Vis spectra of samples incubated with pentenoic or hexenoic acid were unaffected by a desalting step, indicating that pentenoic and hexenoic acid irreversibly binds to *Ta*Fdc/*Ta*FdcV. Quantitative GC assay indicates that pentene production from hexenoic acid by *An*Fdc is limited to a single turnover (Supplementary Fig. 4).

**Crystal structures of *Ta*Fdc and *Ta*FdcV reveal mutation impact on the substrate-binding pocket**. In order to understand how *Ta*FdcV mutations aid in isobutene production, crystal structures of *Ta*Fdc and *Ta*FdcV were solved at a resolution of 1.74 and 1.89 Å, respectively. An overlay of the wild-type and the variant crystal structures shows that the key residues F447, Q200 and the catalytic network of E287–R183–E292 are unaffected by the mutations (Fig. 2A)[16]. The T405M mutation is located at the active site, extending towards the space above the prFMN uracil ring while the Q448W and F404Y mutations are situated in the second shell from the active site. The E292 residue side chain occupies 'up' and 'down' conformations, while weak electron density suggests a high degree of mobility for the L449 side chain. The mobile E292 and L449 gate access to the active site (Fig. 2B) while the Q448W mutation in *Ta*FdcV narrows the binding pocket (Fig. 2C). The T405M and Q448W mutations are likely to be responsible for the increased selectivity for 3-methylcrotonic acid in *Ta*FdcV by enhancing the substrate/active site shape complementarity, blocking access to larger substrates (Supplementary Fig. 5). While comparison of *Ta*Fdc and *Ta*FdcV crystal structures reveals the basis for increased selectivity in the evolved enzyme, it is not immediately clear why 3-methylcrotonic acid can yield isobutene from **Int3**.

**Formation of stable cycloadducts with inhibitors**. The effects of crotonic and 2-butynoic acid on *Ta*FdcV were studied to determine whether the mutations that increase in 3-methylcrotonic acid turnover also affected activity with related compounds. Incubation of *Ta*FdcV with 2-butynoic and crotonic acid led to the familiar split of the 380 nm prFMN peak in the UV–Vis spectrum (Fig. 3A and D), similar to 3-methylcrotonic acid. However, as previously observed with pentenoic and hexenoic acid, the spectrum did not recover the following desalting, suggesting that a covalent inhibitory adduct is formed. Similar trends were observed with *Ta*Fdc, however, incubation at higher inhibitor concentration was required to drive changes in the UV–Vis spectrum.

Upon addition of crotonic acid, a gradual shift in UV–Vis spectrum occurs over minutes, allowing estimation of adduct formation rate (Supplementary Fig. 6A). The observed rate remains first order with respect to crotonic acid, with $k_{obs} = 0.34 \pm 0.03$ min$^{-1}$ at the highest concentration tested (50 mM)

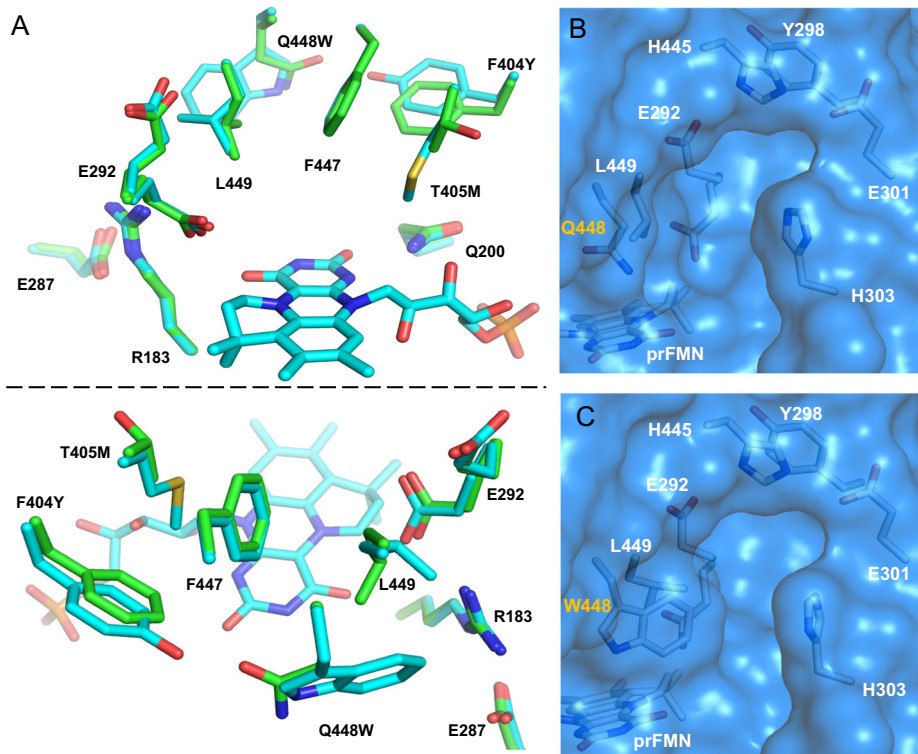

**Fig. 2 Comparison of *Ta*Fdc and *Ta*FdcV. A** Two views of the overlay of *Ta*Fdc wild-type (green, 7NEY [10.2210/pdb7NEY/pdb]) and *Ta*FdcV (blue, 7NF0 [10.2210/pdb7NF0/pdb]) active sites. Comparison of *Ta*Fdc **B** and *Ta*FdcV **C** binding pockets. The mobile L449 and E292 gate the entrance to the active site and the Q448W mutation narrows the entrance to the active site.

(Supplementary Fig. 6B). In contrast, a similar shift in UV–Vis spectrum upon addition of the substrate 3-methylcrotonic acid occurs rapidly within seconds, and at substantially lower 3-methylcrotonic acid concentrations. This suggests that crotonic acid adduct formation is hindered by a higher $K_D$ and/or slower rate of cycloaddition.

ESI–MS and co-crystallization studies confirmed that the *Ta*FdcV 2-butynoic acid adduct stalls as **Int1**, while the *Ta*FdcV crotonic acid adduct undergoes decarboxylation to stall at the **Int3** species (Fig. 3). Similar behaviour has been reported for *An*Fdc[17]. No decarboxylation of the **Int1** with 2-butynoic acid was detected, even in 1-month-old crystals. In contrast, although only **Int3** was observed in co-crystals with crotonic acid, ESI–MS also showed a peak for the corresponding **Int1** (Fig. 3E). It is unclear whether **Int1** can be detected in this case because decarboxylation of crotonic acid is slow, or because there is an equilibrium between **Int1** and **Int3** at ambient $CO_2$ levels.

**An*Fdc*II with three point-mutations has an identical active site conformation to *Ta*FdcV.** To further understand how the architecture of the active site affects the decarboxylation of 3-methylcrotonic acid, corresponding key mutations from *Ta*FdcV were introduced in *An*Fdc. *An*Fdc has been established as a model system due to the fact that it readily yields atomic resolution crystal structures[11,16–18]. Two variants were studied: *An*Fdc T395M (*An*FdcI) and the triple mutant *An*Fdc T395M R435P P438W (*An*FdcII). Overlay of the *An*Fdc wild-type and *Ta*FdcV crystal structures reveals a downward shift of the Y404 residue in *Ta*FdcV in the secondary shell compared to the corresponding Y394 in *An*Fdc (Fig. 4A). The Y394 residue is unaffected in the *An*FdcI variant compared to wild-type (Fig. 4B). In contrast, the active site of the *An*FdcII variant matches that of *Ta*FdcV in the conformation of Y394 and M395 (Fig. 4C).

As expected, neither *An*FdcI nor *An*FdcII were active with cinnamic acid, likely due to a clash between the substrate phenyl ring and M395. While binding of crotonic acid in *An*Fdc wild-type cannot be detected by the UV–Vis spectra over 2-h incubation, both mutants *An*FdcI and *An*FdcII readily bind the inhibitor, evident from UV–Vis spectra, demonstrating increased selectivity towards smaller substrates.

**Ta*FdcII* has comparable isobutene production activity to *Ta*FdcV.** Selected *Ta*Fdc variants (wild-type, *Ta*FdcI i.e. T405M, *Ta*FdcV, *Ta*FdcII, see Supplementary Fig. 1) and *An*Fdc variants (wild-type, *An*FdcI, *An*FdcII) were purified and assayed for isobutene production. *Ta*FdcII (F404Y, T405M, V445P, Q448W) was created by rational design based on the structural analysis of *Ta*Fdc wild-type, *Ta*FdcV and *An*FdcII. A comparison of the isobutene titre obtained following 2 and 4 h incubation revealed *Ta*FdcI and *An*FdcI produced 4–9 times the amount of isobutene compared to the wild-type enzymes. Additional mutations in *An*FdcII and *Ta*FdcII led to a substantial further increase of 18 and 81 fold, respectively, in isobutene production (Supplementary Fig. 7). Surprisingly, the in vitro titer obtained with *Ta*FdcII was slightly higher than the corresponding *Ta*FdcV levels obtained (Fig. 5). Thus, the 4 point mutations in *Ta*FdcII (F404Y, T405M, V445P, Q448W) and 3 point mutations in *An*FdcII (T395M, R435P, P438W), that create an active site architecture identical to *Ta*FdcV (Fig. 4), appear largely responsible for the increased isobutene activity compared to wild-type *Ta*Fdc and *An*Fdc.

While *An*Fdc wild-type was included in the initial UbiD screen, the *An*Fdc wild-type was 90 times lower in activity in vivo compared to *Ta*Fdc. Hence, *An*Fdc was not selected for further directed evolution, despite having comparable in vitro activity to *Ta*Fdc. The disparate and lower activity in vivo might be attributed to *An*Fdc-specific inhibition by metabolites such as phenylacetaldehyde[11]. An initial comparison of in vitro isobutene

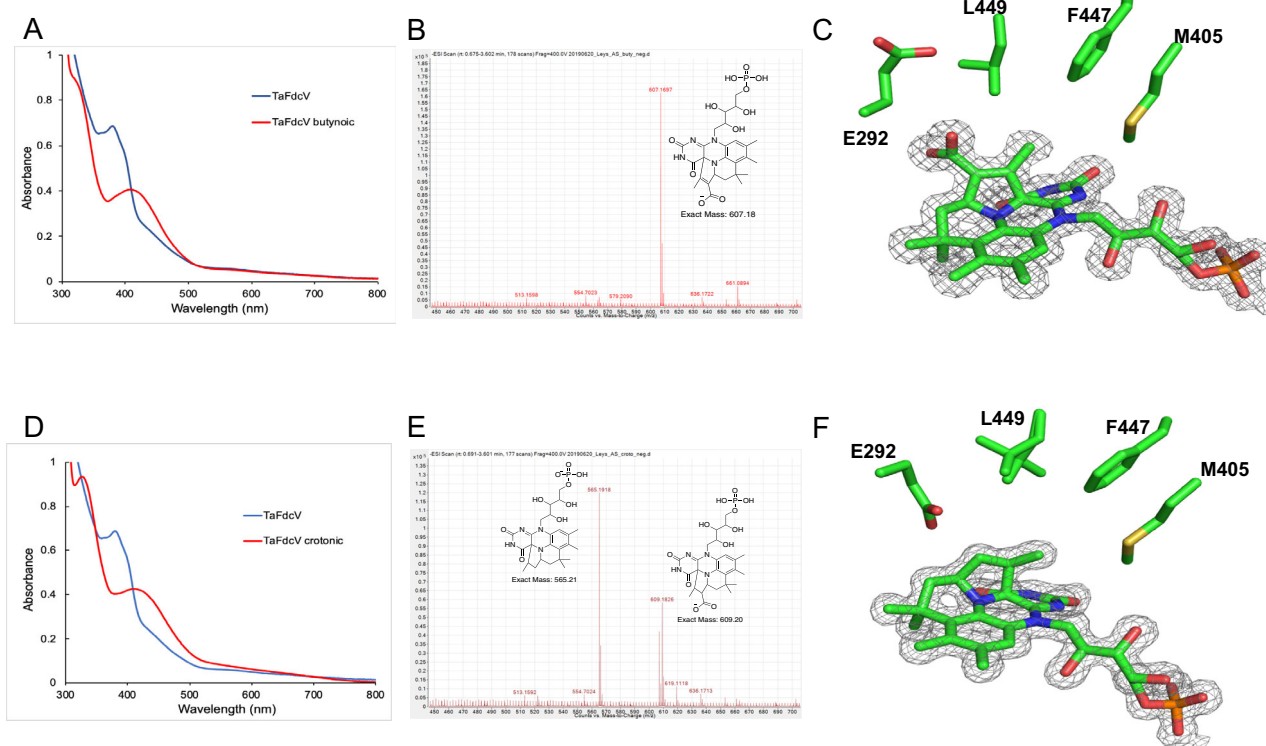

**Fig. 3 Cycloadduct formation in TaFdcV with inhibitors butynoic and crotonic acid. A** UV–Vis spectra of TaFdcV before and after incubating with 2-butynoic acid. **B** ESI–MS spectra of TaFdcV incubated with 2-butynoic acid and desalted, showing the formation of prFMN-butynoic adduct $[M + H]^+ = 607.18$. **C** Crystal structure of TaFdcV with prFMN-butynoic adduct (7NF1 [10.2210/pdb7NF1/pdb]). **D** UV–Vis spectra of TaFdcV as is and incubated with crotonic acid **E** ESI–MS spectra of TaFdcV incubated with crotonic acid (and desalted) showing the formation of decarboxylated prFMN-crotonic adduct $[M–H]^- = 565.21$ and traces of prFMN-crotonic with the carboxylate group $[M–H]^- = 609.20$. **F** Crystal structure of TaFdcV prFMN-crotonic adduct (7NF2 [10.2210/pdb7NF2/pdb]).

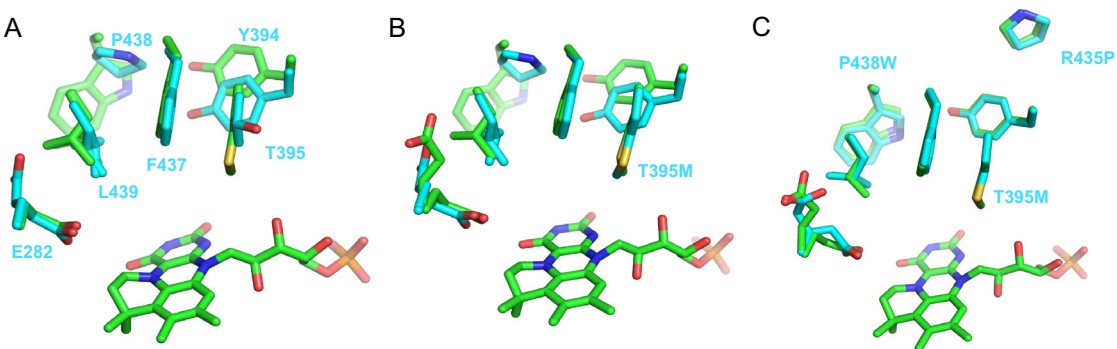

**Fig. 4 Comparison of the TaFdcV and AnFdc wild-type and variant active sites.** Overlay of TaFdcV (green, 7NEY [10.2210/pdb7NEY/pdb]) with **A** AnFdc wild-type (blue, 4ZA4 [10.2210/pdb4ZA4/pdb]), **B** AnFdcI (blue, 7NF3 [10.2210/pdb7NF3/pdb]) and **C** AnFdcII (blue, 7NF4 [10.2210/pdb7NF4/pdb]) crystal structures (AnFdc variant residue numbering according to AnFdc).

production levels using crude cell lysate from cells expressing TaFdc variants with those expressing MVD and/or M3K reveals a ~50-fold increase is observed for TaFdcV compared to MVD/M3K levels (Supplementary Fig. 8). This demonstrates that the evolved TaFdcV is vastly superior in catalysing the decarboxylative step compared to the previously described enzyme systems.

**Computational studies reveal a mechanistic basis for isobutene production.** It is curious that a single methyl group difference, as occurs between crotonic acid and 3-methylcrotonic acid, determines whether the compound is a substrate or inhibitor for the evolved Fdc variants. The marked influence of the additional

methyl group on **Int3** cycloelimination suggests this step may proceed via a cationic or radical beta carbon stabilized through additional hyperconjugation effects. A density functional theory (DFT) active site 'cluster' model (Supplementary Fig. 9) was used to investigate why 3-methylcrotonic acid is decarboxylated and eliminated by TaFdcV in contrast to crotonic acid.

The potential energy surface for the cycloelimination of **Int3** to the non-covalent product complex was computed for both crotonic acid and 3-methylcrotonic acid by varying the $C_\alpha$–$C_{1'}$ and $C_\beta$–$C_{4a}$ bond lengths (Fig. 6). These suggest that 3-methylcrotonic acid undergoes a more asynchronous elimination, with the transition state $C_\alpha$–$C_{1'}$ and $C_\beta$–$C_{4a}$ bond lengths of 1.96 and 2.97 Å, respectively, compared to 1.95 and 2.77 Å for crotonic acid,

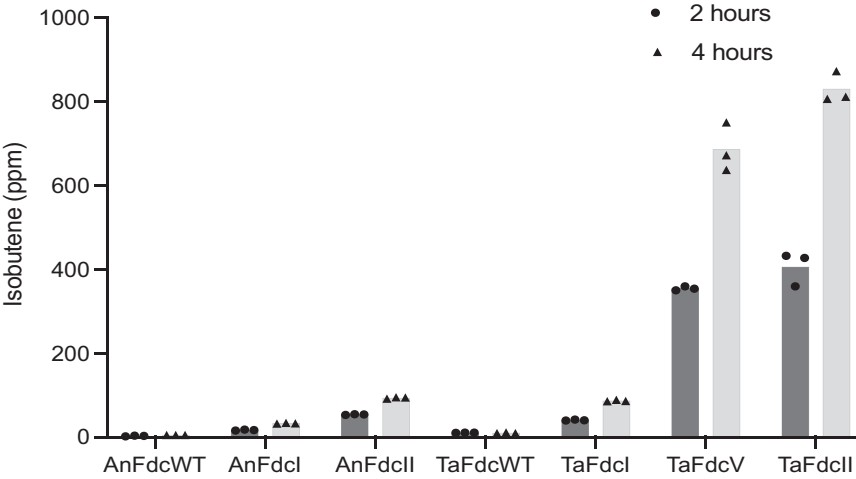

**Fig. 5 Isobutene production by *Ta*Fdc and *An*Fdc variants.** 3-methylcrotonate decarboxylation assay with purified enzyme comparing isobutene production as detected by GC by *Ta*Fdc and *An*Fdc variants (Table 1), both with N-terminal His-tags, over 2 and 4 h with 10 mM 3-methylcrotonate and 0.3 mg/mL enzyme. Fold increase comparison is shown in Supplementary Fig. 7. Source data are provided as a Source Data file.

respectively. This is linked to an increased charge separation occurring between the $C_\beta$ and prFMN for the 3-methylcrotonic acid compared to crotonic acid (Supplementary Tables 4 and 5), possibly affected by additional hyperconjugation in the case of 3-methylcrotonic acid. The release of propene from crotonic acid **Int3** cycloadduct is more endothermic by ~8 kJ mol$^{-1}$ and has a higher energy barrier by 19 kJ mol$^{-1}$ compared to the release of isobutene from **Int3** with 3-methylcrotonic acid. If the activation entropy is similar for the two reactions then the transition state energy difference translates to a ~2200 slower rate for the release of propene from **Int3** at 293 K, explaining the lack of crotonic acid turnover under conditions tested.

**The limit of prFMN-dependent (de)carboxylation by UbiD enzymes.** Directed evolution of *Ta*Fdc to *Ta*FdcV resulted in a marked increase in activity with 3-methylcrotonic acid. Surprisingly, the evolved mutant remained unable to convert crotonic acid to the corresponding propene. This contrasts with previous evolved studies aimed at expanding the *An*Fdc substrate repertoire to include (hetero)aromatic compounds[18]. In this case, the evolution of activity against heteroaromatic bicyclic compounds yielded a broad specificity variant able to convert even naphthoic acid. It is thus possible that 3-methylcrotonic acid represents a limit for bona fide UbiD-substrates, indicating that prFMN-dependent catalysis requires more than an α,β-unsaturated acrylic acid (i.e. a secondary $C_\beta$ carbon) to yield reversible cycloelimination. Indeed, crotonic acid readily forms irreversible adducts with (evolved) Fdc that proceed to the last step prior to product formation. Detailed studies of the *An*Fdc mechanism revealed considerable enzyme-induced strain in substrate-cofactor adducts that avoid dead-end local energy minima during the covalent catalysis[17]. In the case of smaller substrates such as (3-methyl) crotonic acid, the scope for enzyme-induced strain as a tool to optimize the energy landscape is minimal. In this case, cycloelimination of isobutene appears feasible at ambient conditions whereas propene production is not. Computational studies provide a rationale behind these observations, suggesting a ~2200 fold slower rate for the release of propene from **Int3**. Thus, further optimization of isobutene production and future evolution of propene producing Fdc variants will need to focus on the energetics of the hydrocarbon elimination step.

## Methods

**In vivo isobutene assay.** In vivo screenings were carried out on a 96-well plate (DW96, 2.2 mL wells, sealed with a foil sheet). *Ta*Fdc and other UbiD homologues were co-expressed with UbiX (*E. coli*, K-12) in a petDuet vector (UbiD in MCS1 and UbiX in MCS2) in *E. coli* (BL21, DE3). Isobutene production from 0.4 mL reaction mix with 10 mM 3-methylcrotonic acid was detected from the headspace by gas chromatography. The GC method consisted of 100 μL of headspace with a split ratio of 10 injected to RTX-1 column (15 m, 0.32 mm internal diameter, 5 μm film thickness, from RESTEK 10178-111) using nitrogen as a carrier gas (1 mL/min flow rate). The oven temperature was held at 100 °C and the injector and detector were maintained at 250 °C. Isobutene was calibrated at 1000, 5000 and 10,000 ppm with standards from Messer.

**Mutagenesis.** Point mutations (*Ta*FdcI, *Ta*FdcII, *An*FdcI, *An*FdcII, Supplementary Table 6) were generated with a Q5 mutagenesis kit from New England Biolabs. Primers were designed with NEBaseChanger (New England Biolabs). The presence of the point mutation was confirmed by sequencing (Eurofins).

**Protein expression.** A pETDuet-1 vector containing genes for *T. atroviride* Fdc (with an N-terminal 6-histidine affinity tag) and UbiX (*E. coli*, K-12) was transformed into BL21(DE3) competent cells following the manufacturer's protocol (Novagen). A colony was inoculated into Lysogeny Broth (supplemented with 100 μg/mL ampicillin) and incubated by shaking overnight at 37 °C. 5 mL of LB culture was inoculated into 1 L of Terrific Broth (TB, Formedium), supplemented with 100 μg/mL ampicillin. The culture was incubated by shaking at 37 °C until the optical density of 0.6–0.8. The cells were induced with 0.4 mM isopropyl β-D-1-thiogalactopyranoside (IPTG) and supplemented with 0.5 mM MnCl$_2$. The cultures were incubated by shaking at 18 °C for 24 h. The cells were harvested by centrifugation (10 min, 8939 × g) and frozen.

**Protein purification.** Frozen cells were supplemented with EDTA-free protease inhibitor mixture (Roche Applied Science), lysozyme, DNAse, and RNAse (Sigma) and resuspended (50 mM HEPES, 300 mM KCl, pH 6.8). The cells were lysed by sonication on ice (Bandelin Sonoplus sonicator, TT13/F2 tip, 30% power with 20 s on/40 s off for 15 min) and centrifuged (1 h, 174,000–185,500 × g, Beckman Optima LE-80k ultracentrifuge, Ti50.2 rotor). The cell-free extract was loaded on Ni-NTA resin, washed with 4 column volumes of 40 mM imidazole buffer (40 mM imidazole, 50 mM HEPES, 300 mM KCl, pH 6.8) and eluted in 1 mL fractions with 250 mM imidazole buffer (250 mM imidazole, 50 mM HEPES, 300 mM KCl, pH 6.8). Fractions containing protein were combined and desalted into 25 mM HEPES, 150 mM KCl, pH 6.8. Exposure to light was minimized by covering with foil and using black Eppendorf tubes.

**Cycloadduct formation.** *Ta*FdcV (500 μL, 0.45 mM protein, 25 mM HEPES, 150 mM KCl, pH 6.8) was incubated with crotonic or 2-butynoic acid. The formation of the prFMN-crotonic acid cycloadduct was followed by UV–Vis spectroscopy and additional acid were added until full conversion (complete loss of the 380 nm peak). The protein was desalted to 25 mM HEPES, 150 mM KCl, pH 6.8 and plated for crystal trials.

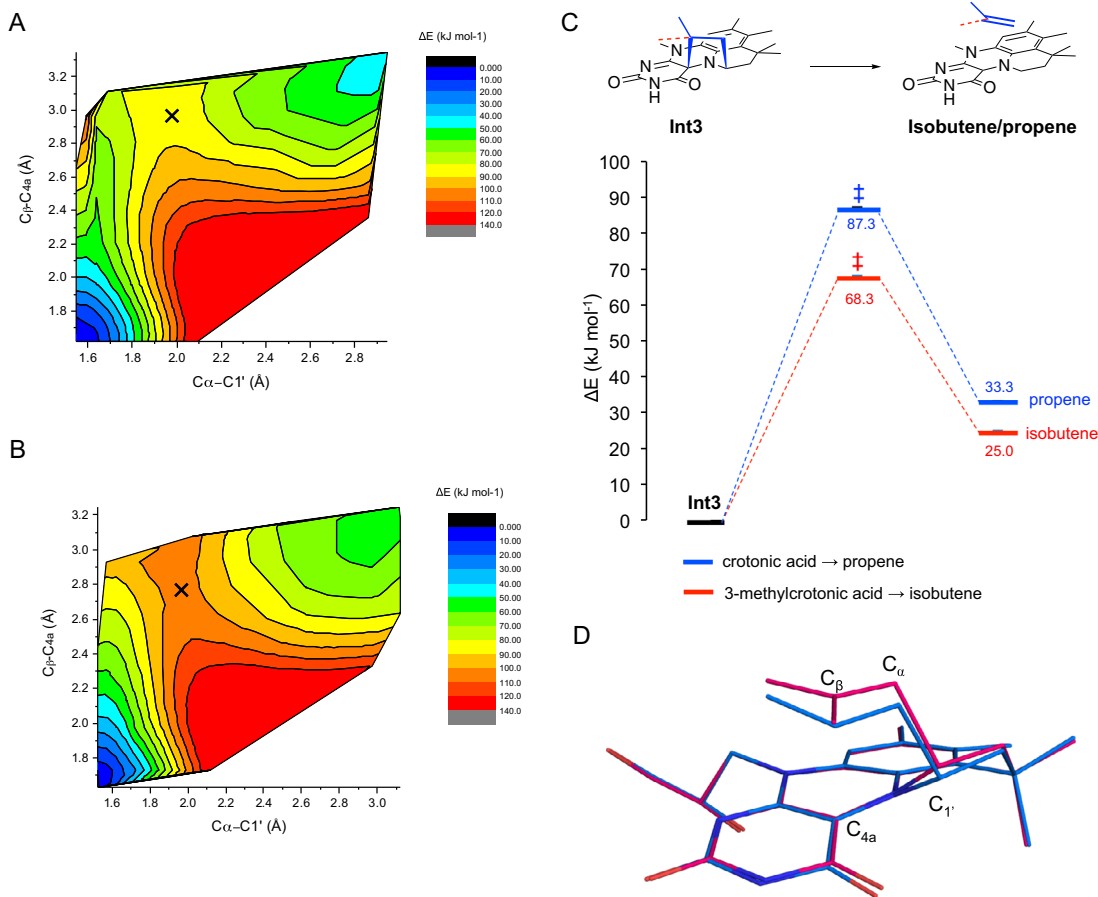

**Fig. 6 DFT calculations applied to the active site of *Ta*Fdc with 3-methylcrotonic and crotonic acid.** Contour map of the potential energy (kJ mol$^{-1}$) landscape for 3-methylcrotonic acid (**A**) and crotonic acid (**B**) conversion to isobutene and propene, respectively, from **Int3** by *Ta*FdcV, projected transition state is marked by X. **C** Zero-point energy corrected potential energy (kJ mol$^{-1}$, Supplementary Tables 2 and 3) scheme for 3-methylcrotonic (red) and crotonic acid (blue) with the **Int3** set as 0 and the projected approximate transition state denoted with double daggers. **D** overlay of the DFT optimized transition states between **Int3** and product for 3-methylcrotonic (pink, $C_\alpha$–$C_{1'}$ and $C_\beta$–$C_{4a}$ bond lengths of 1.96 and 2.97 Å, respectively) and crotonic acid (blue, $C_\alpha$–$C_{1'}$ and $C_\beta$–$C_{4a}$ bond lengths of 1.95 and 2.77 Å, respectively). Source data underlying **A**–**C** are provided as a Source Data file.

**Crystallization and X-ray structure determination.** Crystallization was performed by sitting-drop vapour diffusion. Screening of 0.3 μL of 1 mg/mL *Ta*FdcV in 25 mM HEPES, 150 mM KCl, pH 6.8, and 0.3 μL of reservoir solution at 4 °C resulted in a number of hits in the BCS plate from molecular dimensions. Seed stocks were used to reproduce *Ta*Fdc wild-type crystals and co-crystals with 2-butynoic and crotonic acids in the BCS plate. Crystals were cryoprotected with PEG200 and flash-frozen in liquid nitrogen. *An*Fdc wild-type and variants were crystallized in 0.2 M potassium thiocyanate, Bis–Tris propane 6.5, 20% w/v PEG 3350 at 4 °C[16]. Diffraction data were collected at Diamond beamlines and processed using the CCP4[21] suite version 7.1 (Supplementary Table 7). Phaser MR version 2.8.3 was used to perform molecular replacement using 4ZA4 [https://doi.org/10.2210/pdb4ZA4/pdb] as a model. Refinement was carried out with REFMAC5[22] and manual rebuilding in COOT[23] version 0.9.5. Ligand definitions and coordinates were generated with AceDRG[24].

**In vitro isobutene assay comparing *Ta*Fdc and *An*Fdc variants.** All variants were grown in BL21(DE3) cells with a pETDuet plasmid with the Fdc variant (N-terminal 6-His-tag) in the first multiple cloning site and UbiX (untagged, wild-type from *E. coli* K12) in second multiple cloning site. The cells were grown in a ZYM-5052 auto-inducing medium (30 °C for 6 h, followed by 18 °C for 24 h). The Fdc variants were purified with Protino® Ni-IDA column and stored at −80 °C (in 50 mM Tris–HCl pH 7.5, 1 mM MnCl$_2$, 20 mM NaCl, 200 mM KCl, 10% glycerol). Decarboxylation of 3-methylcrotonate was set up in triplicates in 50 mM Tris–HCl pH 7.5, 1 mM MnCl$_2$, 20 mM NaCl, 200 mM KCl with 10 mM 3-methylcrotonate and 0.3 mg/mL enzyme in DW384 plates (40 μL per well, sealed with foil sheet). Isobutene production was measured from headspace by gas chromatography after 2 and 4 h.

**In vitro isobutene assay comparing *Ta*Fdc, *Sc*MVD and *Pt*M3K.** An equal amount of *E. coli* cells containing either empty pETDuet (as control) or one of the following plasmids: pETDuet TaFdc_UbiX, pETDuet TaFdcV_UbiX, pETDuet

PtM3K (*P. torridus* mevalonate 3-kinase, Uniprot: Q6KZB1), pETDuet ScMVD (*S. cerevisiae* MVD, Uniprot: P32377) or pETDuet PtM3K—ScMVD, were lysed in 50 mM Tris–HCl pH 7.5, 20 mM KCl, 2 mM MgCl$_2$, 1 g/L lysozyme, 0.03 g/L DNAse for 1 h at 37 °C. A total of 150 μL of lysate was transferred to a 2 mL GC-vial and MgCl$_2$ (10 mM final concentration) was added. Substrates were added to 50 mM final concentration and 200 μL total volume, and consisted of either 3-hydroxyisovalerate/ATP, 3-phosphonooxy-isovalerate/ADP or 3-methylcrotonate. Following 4 h of incubation at either 37 or 50 °C, the reaction mixture was inactivated by incubation at 90 °C for 5 min. GC analysis of the gas phase was carried out as described above to determine isobutene levels produced. All reactions were carried out in duplicates.

**DFT calculations.** *Ta*FdcV active site cluster model with crotonic (365 atoms) and 3-methylcrotonic acid (368 atoms) was built based on the *Ta*FdcV crystal structure with crotonic acid bound as **Int3** adduct (Supplementary Fig. 9) and modelled at the B3LYP/6–31 G(*d*,*p*) level of theory with the D3 version of Grimme's dispersion with Becke–Johnson damping and a generic polarizable continuum with $\varepsilon = 5.7$ using the polarizable continuum model[25]. $C_\alpha$–$C_{1'}$ and $C_\beta$–$C_{4a}$ bonds were both fixed for any single DFT optimization and substrate release was modelled using Gaussian 09 revision D.01. by lengthening one bond by 0.05 Å at a time, resulting in a 3D energy landscape consisting of ~900 DFT optimized models (for 3-methylcrotonic acid).

**Reporting summary.** Further information on research design is available in the Nature Research Reporting Summary linked to this article.

## Data availability
Data supporting the findings of this work are available within the paper and its Supplementary Information files. A reporting summary for this Article is available as a Supplementary Information file. Crystal structure data that support the findings of this

study have been deposited in the PDB with the accession codes 7NEY, 7NF0, 7NF1, 7NF2, 7NF3 and 7NF4. Input and output files for DFT calculations are available at Zenodo [https://doi.org/10.5281/zenodo.5137885]. Source data are provided with this paper.

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

## Acknowledgements

This work was supported financially by grant ERC pre-FAB 695013 (D.L.). We acknowledge assistance via use of the Manchester Protein Structure Facility and the Diamond Light Source (proposal nos. MX12788 and MX17773), which contributed to the results presented here. We also acknowledge the assistance given by IT Services and the use of the Computational Shared Facility at The University of Manchester. D.L. is a Royal Society Wolfson Merit Award holder.

## Author contributions

B.V., F.S., M.A. and M.C. performed initial in vivo screening for isobutene activity, directed evolution of *Ta*Fdc and isobutene assays with *An*Fdc and *Ta*Fdc variants, including expression and purification. A.S. performed point mutagenesis, enzyme expression, purification and characterization by UV–Vis absorption, inhibition studies and the assay with hexenoic acid. A.S. performed crystallization and determined crystal structures with assistance from D.L., R.S. assisted with mass spectrometry. A.S. and S.H. performed computational studies. A.S. and D.L. wrote the initial draft of the manuscript. All authors discussed the results and commented on the manuscript. D.L. and M.A. initiated and coordinated the project.

## Competing interests

B.V., F.S., M.A. and M.C. are employees of Global Bioenergies. Other authors declare no competing interests.
