## [Peer Review File · Nature Communications]

Directed evolution of prenylated FMN-dependent Fdc supports efficient in vivo isobutene productionREVIEWER COMMENTS

Reviewer #1 (Remarks to the Author):

The manuscript "Directed evolution of prFMN-dependent Fdc supports efficient in vivo isobutene production" by Saaret A. et al. is the latest report from the research group that has discovered prFMN for the first time as a novel flavin coenzyme. The authors performed directed evolution of ferulic acid decarboxylase from a fungus *Trichoderma atroviride* (TaFdc), by in vivo enzyme screening on *E. coli* system that allows the selection of TaFdc from 15 homologs and following four rounds of mutagenesis, while details of the process are not described. The resulting mutant TaFdcV efficiently catalyzes decarboxylation of 3-methylcrotonic acid with much higher activity than that of wild type TaFdc, losing activity toward cinnamic acid, a good substrate for the wild type. This substrate specificity was explained by the comparison of the crystal structures of the mutant and wild type TaFdc, and the change in substrate specificity by mutagenesis was reproduced with homologous Fdc from *Aspergillus niger*. Moreover, the analyses of the cycloadducts between the cofactor prFMN and substrate/inhibitor/intermediate revealed the limit of prFMN-dependent decarboxylase as for the structure of substrates. The fact that Fdc can catalyze decarboxylation of 3-methylcrotonic acid but cannot that of crotonic acid suggested that the rate-limiting cycloelimination step proceeds via cationic (or radical) mechanism. This hypothesis was supported in a clear way by computational study with DFT calculations based on TaFdc structures.

The authors made an excellent job to create valuable mutant enzymes for future isobutene bio-production. They also made important discoveries about the catalytic mechanism of prFMN-dependent decarboxylase. All the data in the manuscript look very sound and persuasive. The reviewer only requires minor modification of the manuscript as listed below:

- 1) In the title, the authors should use not abbreviations but common words. An example of an appropriate title is "Directed evolution of prenylated FMN-dependent ferulic acid decarboxylase supports efficient in vivo isobutene production."
- 2) The description "conjugated acrylic acid" seems redundant because acrylic acid itself has conjugated C=O and C=C double bonds. Similarly, "non-conjugated acrylic acid" is unacceptable. The reviewer recommends the use of "acrylic acid with (or without) extended conjugation" instead.
- 3) At the first appearance, the name of microorganism should not be abbreviated, like *A. niger* in page 2 and *T. atroviride* in page 4.
- 4) In page 2: "Following CO₂ for E282 exchange," should be "Following the exchange of CO₂ with E282,".
- 5) In page 4, the authors describe that "the equivalent mutation (to TaFdcV) supported significant increase in isobutene production when introduced in four other Fdc homologues". This data should be added as supplementary information.
- 6) The way of directed mutagenesis should be described in more detail. Why and how saturation mutagenesis was performed in the 1st and 3rd rounds? How many mutants were screened in the 2nd and 4th rounds to select the mutants? If it is a long story, please add it as supplementary information.
- 7) In page 5, line 5: MS-ESI > ESI-MS
- 8) In page 5, line 16: no long-lived > non-long lived
- 9) In page 6, line 9: where > were
- 10) In page 6: Was "product extraction for pentenoic and hexenoic acid substrates" performed in a similar way with the extraction of cycloadduct, which is described in the Methods section? The experimental methods for the various analyses using pentenoic and hexenoic acid should be described in detail as supplementary information.
- 11) In page 7: "Butynoic acid" should be "2-butynoic acid" for clarity.
- 12) Throughout the manuscript: The microorganism name in the abbreviations such as "Ta" in TaFdc is italicized in some parts in the manuscript but not italicized in other parts. Please unify them.
- 13) In page 12: "An unsaturated acrylic acid" should be "an α,β -unsaturated carboxylic acid" or just "an acrylic acid moiety" because acrylic acid is originally unsaturated.
- 14) In the legend of Figure 6: "DFT optimized Int3 to product transition states" > "DFT optimized transition states between Int3 and product"

- 15) In pages 14 and 16: How was the headspace gas sampling performed? Were deep well plates sealed throughout reaction? If so, what was used for sealing? Please describe in the Methods section.
- 16) In page 15: Although the authors described "the resultant yellow solid was added to apo-Fdc (*A. niger*)", the apo-Fdc used is considered to be TaFdcV from Table 1 and the legend of Figure 3. In addition, please describe how the apo-TaFdcV was prepared.
- 17) In page 16, line 8: molecular placement > molecular replacement
- 18) In Figure S1: Panels H, I, and J are not referred in the main text. Can they be removed?
- 19) Figure S4 is referred in the main text faster than S2 and S3. Please change the numberings in good order.

Reviewer #2 (Remarks to the Author):

In this excellent manuscript, the Authors describe their successful efforts to develop a mutant of ferulic acid decarboxylase (Fdc) that is able to produce isobutene from 3-methylcrotonic acid. Fdc is by now widely known as the prototype of the enzymes that use prenylated flavins to afford the decarboxylation of highly conjugated substrates. This work now shows that Fdc enzymes can also work on non-aromatic/non-conjugated substrates such 3-methylcrotonic acid. Curiously, the Authors found that a substituent on the 3 position is essential for catalysis. Rather than being a substrate, crotonic acid is indeed found to be an inhibitor by forming a stable adduct with the prenylated flavin. DFT calculations indicate that the 3-methyl group creates the proper charge distribution and separation on the intermediate atoms as requested for intermediate decay and concomitant release of the decarboxylated product. This is very good work with an impact in the field of biocatalysis as well as in basic enzymology. Isobutene is a valuable compound and a biocatalytic route for its production is much needed. At the same time, this work provides considerable insight into the prenylated-favins and their reactivities, demonstrating that their substrate scopes can be considerably expanded by mutagenesis and protein discovery through genome mining. However, some limitations are posed by the necessity to avoid formation of "too stable dead-end" covalent intermediates. The manuscript is very well written. The Authors employed several analytical, enzymological, and structural techniques to support their conclusions. I have a few comments, mainly about a few points that should be clarified.

-The Authors mention the detrimental effect of the histidine-tag on the activities. Can they elaborate on this observation? Any hypothesis? Here, I am actually confused because Figure S3 reports the conversions measured with the His-tagged proteins. The data for the untagged proteins should be shown as well. Overall, this point must be clarified.

-The structural superposition of Figure 2A should be shown in two orientations to allow the reader to fully appreciate the amino acid replacements and associated minor structural changes.

-The text of page 10 suddenly mentions the TaFdcI and TaFdcII variants without any previous description.

-Would it be possible to include a supplementary or main text picture showing the binding of the substrates in the active site as it can be inferred by simple docking and previous structural work? The reader would like to see how 3-methylcrotonic acid and ferulic acid are known and/or expected to interact with the mutagenized side chains.

Andrea Mattevi

Reviewer #3 (Remarks to the Author):

General comments:

The manuscript by Saaret et al., is a well written paper on a modified mevalonate pathway using an

evolved ferulic acid decarboxylase enzyme. This enzyme catalyzes a reversible non-oxidative decarboxylation of 3-methylcrotonic acid to isobutene. Not only does this work provide another pathway for isobutene production that is potentially more economical, significant work was performed on enzyme structure and function that significantly advances our understanding of the enzyme kinetics. The only major comment that I have does not deter from the significance of the manuscript. It would be nice in the results and discussion to have a more direct comparison to the previous literature. Currently, there is no way for me to compare if this enzyme is capable of producing higher titers faster than the M3K or MVD enzymes. In this respect it would be good to normalize the production per cell per time. It would also be helpful to switch Figure 5 for Figure S3 and have the text refer to the fold increase.

Specific comments:

- Spell out genus name the first time it is used then abbreviate.
- Please add a few more details on the GC- i.e. column, standards. Was this paired with a mass spec?
- Please add the organism for each enzyme in Table S1.

Brandon Briggs

Point-by-point response to authors comments:

Reviewer #1 (Remarks to the Author):

The manuscript "Directed evolution of prFMN-dependent Fdc supports efficient in vivo isobutene production" by Saaret A. et al. is the latest report from the research group that has discovered prFMN for the first time as a novel flavin coenzyme. The authors performed directed evolution of ferulic acid decarboxylase from a fungus *Trichoderma atroviride* (TaFdc), by in vivo enzyme screening on *E. coli* system that allows the selection of TaFdc from 15 homologs and following four rounds of mutagenesis, while details of the process are not described. The resulting mutant TaFdcV efficiently catalyzes decarboxylation of 3-methylcrotonic acid with much higher activity than that of wild type TaFdc, losing activity toward cinnamic acid, a good substrate for the wild type. This substrate specificity was explained by the comparison of the crystal structures of the mutant and wild type TaFdc, and the change in substrate specificity by mutagenesis was reproduced with homologous Fdc from *Aspergillus niger*. Moreover, the analyses of the cycloadducts between the cofactor prFMN and substrate/inhibitor/intermediate revealed the limit of prFMN-dependent decarboxylase as for the structure of substrates. The fact that Fdc can catalyze decarboxylation of 3-methylcrotonic acid but cannot that of crotonic acid suggested that the rate-limiting cycloelimination step proceeds via cationic (or radical) mechanism. This hypothesis was supported in a clear way by computational study with DFT calculations based on TaFdc structures.

The authors made an excellent job to create valuable mutant enzymes for future isobutene bio-production. They also made important discoveries about the catalytic mechanism of prFMN-dependent decarboxylase. All the data in the manuscript look very sound and persuasive. The reviewer only requires minor modification of the manuscript as listed below:

1) In the title, the authors should use not abbreviations but common words. An example of an appropriate title is "Directed evolution of prenylated FMN-dependent ferulic acid decarboxylase supports efficient in vivo isobutene production."

This has now been fixed

2) The description "conjugated acrylic acid" seems redundant because acrylic acid itself has conjugated C=O and C=C double bonds. Similarly, "non-conjugated acrylic acid" is unacceptable. The reviewer recommends the use of "acrylic acid with (or without) extended conjugation" instead. **This has now been fixed**

3) At the first appearance, the name of microorganism should not be abbreviated, like *A. niger* in page 2 and *T. atroviride* in page 4.

This has now been fixed

4) In page 2: "Following CO₂ for E282 exchange," should be "Following the exchange of CO₂ with E282,".

This has now been fixed

5) In page 4, the authors describe that "the equivalent mutation (to TaFdcV) supported significant increase in isobutene production when introduced in four other Fdc homologues". This data should be added as supplementary information. **We have removed this sentence from the manuscript.**

- 6) The way of directed mutagenesis should be described in more detail. Why and how saturation mutagenesis was performed in the 1st and 3rd rounds? How many mutants were screened in the 2nd and 4th rounds to select the mutants? If it is a long story, please add it as supplementary information. **Figure S1 added to supplementary material.**
- 7) In page 5, line 5: MS-ESI > ESI-MS **This has now been fixed**
- 8) In page 5, line 16: no long-lived > non-long lived **We have altered the sentence to improve clarity**
- 9) In page 6, line 9: where > were **This has now been fixed**
- 10) In page 6: Was "product extraction for pentenoic and hexenoic acid substrates" performed in a similar way with the extraction of cycloadduct, which is described in the Methods section? The experimental methods for the various analyses using pentenoic and hexenoic acid should be described in detail as supplementary information. **Pentenoic and hexenoic acid adduct were not extracted but the turnover was analysed by MS, we clarified the text.**
- 11) In page 7: "Butynoic acid" should be "2-butynoic acid" for clarity. **This has now been fixed**
- 12) Throughout the manuscript: The microorganism name in the abbreviations such as "Ta" in TaFdc is italicized in some parts in the manuscript but not italicized in other parts. Please unify them. **This has now been fixed**
- 13) In page 12: "An unsaturated acrylic acid" should be "an α,β -unsaturated carboxylic acid" or just "an acrylic acid moiety" because acrylic acid is originally unsaturated. **This has now been fixed**
- 14) In the legend of Figure 6: "DFT optimized Int₃ to product transition states" > "DFT optimized transition states between Int₃ and product" **This has now been fixed**
- 15) In pages 14 and 16: How was the headspace gas sampling performed? Were deep well plates sealed throughout reaction? If so, what was used for sealing? Please describe in the Methods section. **We have added a line to methods section to provide further details.**
- 16) In page 15: Although the authors described "the resultant yellow solid was added to apo-Fdc (*A. niger*)", the apo-Fdc used is considered to be TaFdcV from Table 1 and the legend of Figure 3. In addition, please describe how the apo-TaFdcV was prepared. **This has now been fixed**
- 17) In page 16, line 8: molecular placement > molecular replacement **This has now been fixed**
- 18) In Figure S1: Panels H, I, and J are not referred in the main text. Can they be removed? **Added two sentences about light-sensitivity in text referring to those figures.**
- 19) Figure S4 is referred in the main text faster than S2 and S3. Please change the numberings in good order. **Supplementary figures reordered according to occurrence order in text**

Reviewer #2 (Remarks to the Author):

In this excellent manuscript, the Authors describe their successful efforts to develop a mutant of ferulic acid decarboxylase (Fdc) that is able to produce isobutene from 3-methylcrotonic acid. Fdc is by now widely known as the prototype of the enzymes that use prenylated flavins to afford the decarboxylation of highly

conjugated substrates. This work now shows that Fdc enzymes can also work on non-aromatic/non-conjugated substrates such 3-methylcrotonic acid. Curiously, the Authors found that a substituent on the 3 position is essential for catalysis. Rather than being a substrate, crotonic acid is indeed found to be an inhibitor by forming a stable adduct with the prenylated flavin. DFT calculations indicate that the 3-methyl group creates the proper charge distribution and separation on the intermediate atoms as requested for intermediate decay and concomitant release of the decarboxylated product. This is very good work with an impact in the field of biocatalysis as well as in basic enzymology. Isobutene is a valuable compound and a biocatalytic route for its production is much needed. At the same time, this work provides considerable insight into the prenylated-favins and their reactivities, demonstrating that their substrate scopes can be considerably expanded by mutagenesis and protein discovery through genome mining. However, some limitations are posed by the necessity to avoid formation of “too stable dead-end” covalent intermediates. The manuscript is very well written. The Authors employed several analytical, enzymological, and structural techniques to support their conclusions. I have a few comments, mainly about a few points that should be clarified.

-The Authors mention the detrimental effect of the histidine-tag on the activities. Can they elaborate on this observation? Any hypothesis? Here, I am actually confused because Figure S3 reports the conversions measured with the His-tagged proteins. The data for the untagged proteins should be shown as well. Overall, this point must be clarified. **We have clarified this with figure S7.**

-The structural superposition of Figure 2A should be shown in two orientations to allow the reader to fully appreciate the amino acid replacements and associated minor structural changes. **We have added another view.**

-The text of page 10 suddenly mentions the TaFdcI and TaFdcII variants without any previous description. **We have added an additional sentence.**

-Would it be possible to include a supplementary or main text picture showing the binding of the substrates in the active site as it can be inferred by simple docking and previous structural work? The reader would like to see how 3-methylcrotonic acid and ferulic acid are known and/or expected to interact with the mutagenized side chains. **We have modelled crotonic and 3-methylcrotonic, and added an overlay with alpha-fluorocinnamic (Figure S5).**

Andrea Mattevi

Reviewer #3 (Remarks to the Author):

General comments:

The manuscript by Saaret et al., is a well written paper on a modified mevalonate pathway using an evolved ferulic acid decarboxylase enzyme. This enzyme catalyzes a reversible non-oxidative decarboxylation of 3-

methylcrotonic acid to isobutene. Not only does this work provide another pathway for isobutene production that is potentially more economical, significant work was performed on enzyme structure and function that significantly advances our understanding of the enzyme kinetics. The only major comment that I have does not deter from the significance of the manuscript. It would be nice in the results and discussion to have a more direct comparison to the previous literature. Currently, there is no way for me to compare if this enzyme is capable of producing higher titers faster than the M₃K or MVD enzymes. In this respect it would be good to normalize the production per cell per time. It would also be helpful to switch Figure 5 for Figure S₃ and have the text refer to the fold increase.

We have performed additional experiments and described these in the text and accompanying figure S8.

Specific comments:

- Spell out genus name the first time it is used then abbreviate. **This has now been fixed**
- Please add a few more details on the GC- i.e. column, standards. Was this paired with a mass spec? **We have added some details to the methods sections.**
- Please add the organism for each enzyme in Table S1. **This has now been fixed**

Brandon Briggs

REVIEWERS' COMMENTS

Reviewer #1 (Remarks to the Author):

I am totally satisfied with the revision made by the authors.

Hisashi Hemmi

Reviewer #2 (Remarks to the Author):

The Authors have further improved the manuscript by addressing the comments raised by the Reviewers. Excellent work.

Andrea Mattevi

Reviewer #3 (Remarks to the Author):

All previous comments have been more than adequately addressed. The additional experimentation is greatly appreciated and goes beyond what I was previously suggesting to compare rates from the other enzymes.